# Genetic Testing in Children with Developmental and Epileptic Encephalopathies: A Review of Advances in Epilepsy Genomics

**DOI:** 10.3390/children10030556

**Published:** 2023-03-15

**Authors:** Yu-Tzu Chang, Syuan-Yu Hong, Wei-De Lin, Chien-Heng Lin, Sheng-Shing Lin, Fuu-Jen Tsai, I-Ching Chou

**Affiliations:** 1School of Post Baccalaureate Chinese Medicine, China Medical University, Taichung 40447, Taiwan; zzchang.tw@gmail.com (Y.-T.C.);; 2Division of Pediatric Neurology, China Medical University Children’s Hospital, Taichung 40447, Taiwan; 3Department of Medicine, School of Medicine, China Medical University, Taichung 40447, Taiwan; 4Graduate Institute of Biomedical Sciences, China Medical University, Taichung 40447, Taiwan; 5Department of Medical Research, China Medical University Hospital, Taichung 40447, Taiwan; 6Division of Pediatric Pulmonology, China Medical University Children’s Hospital, Taichung 40447, Taiwan; 7Department of Biomedical Imaging and Radiological Science, College of Medicine, China Medial University, Taichung 40447, Taiwan; 8Division of Genetics and Metabolism, China Medical University Children’s Hospital, Taichung 40447, Taiwan; 9Department of Medical Genetics, China Medical University Hospital, Taichung 40447, Taiwan; 10School of Chinese Medicine, China Medical University, Taichung 40447, Taiwan; 11Department of Medical Laboratory Science and Biotechnology, Asia University, Taichung 40447, Taiwan; 12Graduate Institute of Integrated Medicine, China Medical University, Taichung 40447, Taiwan

**Keywords:** epileptic encephalopathy, genetic testing, developmental delay, whole-genome sequencing, next-generation sequencing

## Abstract

Advances in disease-related gene discovery have led to tremendous innovations in the field of epilepsy genetics. Identification of genetic mutations that cause epileptic encephalopathies has opened new avenues for the development of targeted therapies. Clinical testing using extensive gene panels, exomes, and genomes is currently accessible and has resulted in higher rates of diagnosis and better comprehension of the disease mechanisms underlying the condition. Children with developmental disabilities have a higher risk of developing epilepsy. As our understanding of the mechanisms underlying encephalopathies and epilepsies improves, there may be greater potential to develop innovative therapies tailored to an individual’s genotype. This article provides an overview of the significant progress in epilepsy genomics in recent years, with a focus on developmental and epileptic encephalopathies in children. The aim of this review is to enhance comprehension of the clinical utilization of genetic testing in this particular patient population. The development of effective and precise therapeutic strategies for epileptic encephalopathies may be facilitated by a comprehensive understanding of their molecular pathogenesis.

## 1. Introduction

Developmental and epileptic encephalopathies (DEEs) are a group of enervating neurological conditions that profoundly affect brain development and function. Diagnostic accuracy is crucial for the effective management of DEEs and is generally achieved through a combination of clinical investigations. These investigations may include a thorough review of the patient’s medical history as well as physical examination, neurodevelopmental assessment, and neuroimaging findings. Recently, genetic testing has become increasingly important in the diagnosis and management of DEEs as genetic mutations reportedly play a significant role in numerous cases. A flowchart outlining the DEE investigation process is shown in Figure 1.

## 2. Background of Epilepsy Genetics

Epilepsy, a group of neurological disorders affecting over 50 million people worldwide, has multiple causes, including structural, infectious, metabolic, and immune causes [1]. Genetics plays a significant role in approximately 70% of epilepsy cases, either as a single genetic variant (in rare forms) or as multiple genetic variants combined with environmental factors (in common forms) [2]. Clinical evaluations and genetic testing can provide critical information regarding the underlying genetic factors contributing to epilepsy, allowing for more effective and personalized therapy according to the patient’s unique genetic background. 

Intellectual disability affects 1–3% of the population in Western countries. Children with developmental disabilities are at an increased risk of epilepsy, with a higher prevalence rate than in the general population [3]. Recent research suggests that 70–80% of epilepsy cases have genetic causes. Next-generation sequencing (NGS) technologies, such as targeted gene panels, whole-exome sequencing (WES), and whole-genome sequencing (WGS), have enabled the analysis of hundreds of genes associated with various epilepsy syndromes [4]. The field of epilepsy genetics is expanding and evolving rapidly, driven by technological advances and gene discovery. Epileptic encephalopathies (EEs) are severe forms of childhood epilepsy that are phenotypically heterogeneous with different underlying genetic defects [5,6]. EEs are characterized by refractory seizures and developmental delay and are often accompanied by various psychiatric comorbidities [7,8]. Although EEs are most commonly associated with structural brain defects and inherited metabolic disorders, gene mutations may also play a role in their development, despite no clear genetic inheritance pattern or consanguinity.

To date, approximately 265 genes have been identified in epilepsy; of these, several genes, including *STXBP1, ARX, SLC25A22, KCNQ2, CDKL5, SCN1A*, and *PCDH19*, have been associated with early-onset EEs. Identifying the genetic basis of EEs with developmental delay or intellectual disability is valuable for diagnosing and optimizing anticonvulsant treatment and disease prognosis [9]. The most common EEs include Ohtahara syndrome, early myoclonic encephalopathy, epilepsy of infancy with migrating focal seizures, West syndrome, and Dravet syndrome (DS), which are usually unresponsive to traditional antiepileptic medications. 

## 3. Advances in Disease-Related Gene Discovery

Advances in disease-related gene discovery have led to the identification of new genes or mutations that contribute to the development of specific diseases, such as epilepsy. This area of research has made tremendous strides in recent years owing to advances in genomic technologies, such as WES, WGS, and gene panels. Using these tools, researchers have identified a growing number of disease-related genes, particularly for complex diseases such as EEs. 

The discovery of disease-related genes has also opened new avenues for personalized medicine, in which treatments can be tailored to an individual’s genetic profile, leading to more effective and precise therapies. This is particularly important for children with DEEs, who are often resistant to existing treatments and have a poor prognosis. Overall, advancements in disease-related gene discovery have profoundly impacted our understanding of epilepsy and other diseases, leading to a new era of personalized medicine with the potential to improve the lives of millions of people worldwide.

## 4. Major Advances in Epilepsy Genomics

### 4.1. Clinical Testing including Comprehensive Gene Panels, Exomes, and Genomes

Clinical testing using gene panels, exomes, and genomes has revolutionized the diagnosis of genetic epilepsy. Exome sequencing (ES) covers nearly all coding portions of known genes and is a powerful tool for identifying pathogenic variants in the coding regions of known genes, including those independent of the initial indication for testing (incidental findings). Over the last decade, the arrival of high-throughput sequencing technologies, collectively referred to as NGS or massive parallel sequencing, has greatly reduced sequencing costs and increased speed, leading to a surge in gene discovery for human disorders, and up to 50% monogenic epilepsy cases are now diagnosed with precision [10]. In 2016, EuroGentest published guidelines for diagnostic NGS to help laboratories implement and accredit the use of NGS in the diagnosis of rare diseases, including recommendations for WES [11].

Instead of testing one or a few genes, laboratories can now offer the sequencing of comprehensive gene panels, exomes, or genomes. For ES, analysis of parental samples is recommended, ideally through trio analysis or duo analysis if only one parent is available. This can effectively reduce the number of candidate variants that require review and help with the interpretation of the identified variants [12,13]. Multiple studies have indicated that testing trios, including both probands and their parents, may result in a higher diagnostic yield for ES than sequencing only the proband’s DNA. Research has shown an overall diagnostic yield for epilepsy gene panels of 15–48%, but the results can vary based on the population being tested and the number of genes included in the panel [14,15,16,17]. Although gene panels offer a cost-effective solution, they have some limitations, such as being restricted to the number of genes included in the panel and potentially missing disease-causing variants in unknown genes. The diagnostic yield of gene panels is also dependent on the population being tested and can be as low as 0.8% [18,19]. Target gene panels for neurodevelopmental disabilities (NDDs) can facilitate the identification of mutations in a relatively short time [9]. In contrast, WES offers a more comprehensive approach that covers the entire human coding sequence. This makes WES the preferred method, particularly for the diagnosis of DEEs with greater genetic heterogeneity. The diagnostic success rate of WES is reportedly 25–44% [20,21,22]. WES enables the identification of pathogenic variants, including copy number variants (CNVs), and genetic heterogeneity of de novo variants in NDDs, highlighting trio exome sequencing as an effective diagnostic tool for NDDs [23].

Previous studies indicate that NGS techniques, with their capability for massive parallel sequencing of numerous genes, are efficient and cost-effective diagnostic tools for identifying the genetic causes of EEs [24] (Table 1).

Despite state-of-the-art genetic testing, a significant number of patients with DEEs lack genetic diagnosis. WES is becoming increasingly popular for uncovering the role of noncoding genetic elements in the human genome [25]. Previous studies have recommended genetic testing strategies to achieve the highest clinical value, cost-effectiveness, and diagnostic yield in individuals with epilepsy [20,26,27]. As new tests are introduced and the costs of the existing tests decrease, these testing algorithms are likely to undergo changes. In light of economic and time constraints, the development of targeted gene panels for epilepsy with NDDs may be a viable alternative option [9]. Nevertheless, new assays may be required to address the molecular mechanisms of these less-known but important genes.

In conclusion, the adoption of comprehensive gene panels, exomes, and genomes has significantly increased diagnostic rates and deepened our understanding of the underlying disease processes of DEEs [1]. Given the clinical heterogeneity of gene panels targeting monogenic epilepsies, WES and WGS should be the preferred methods over a single-gene approach that has limited usefulness in epilepsy genetics [28]. The overarching goal is to increase our understanding of the clinical application of genetic testing for DEEs and to develop effective and precise therapeutic strategies based on each child’s unique genetic background.

### 4.2. Impact of Genetic Testing on Diagnostic Rates and Disease Understanding

Genetic testing helps patients and their families understand the underlying causes of their medical conditions, which can assist with the planning of appropriate treatment and management strategies. Chromosomal microarray analysis (CMA) is a powerful tool for detecting clinically significant genomic variants, such as microdeletions and duplications. It can detect changes as small as 5–10 Kb in size, with a resolution up to 1000 times higher than that of conventional karyotyping. This technology is commonly used to identify CNVs that contribute to neurodevelopmental disorders and congenital anomalies. In contrast, WGS and WES can provide more comprehensive and timely diagnoses of genetic diseases. The detection of genomic variants has been greatly enhanced by ES and CMA; however, interpreting their impact on health and development can be challenging.

As a result, some variants may be classified as “variants of uncertain significance” (VUS) based on the available evidence. Studies have shown that patients undergoing CMA for developmental delay, intellectual disability, autism spectrum disorder, or multiple congenital anomalies carry at least one VUS an estimated 7.9–19% of the time [29,30,31,32]. The frequency of VUS in patients undergoing ES varies greatly, with reported rates of 25.3–86% [33,34,35]. The rate of specific VUS is influenced by various factors, such as the laboratory’s reporting practices, phenotypic information provided by the ordering clinician, and the inclusion of parental samples. Nongenetic providers have expressed the need for further education and access to genetics professionals’ expertise to help them understand and disclose these results effectively. It is essential to address these concerns to ensure that patients and their families receive the necessary support and counseling to make informed decisions regarding their healthcare. 

Individualized genetic testing is crucial to optimize the diagnostic yield and cost-effectiveness of genetic testing. This is particularly important, considering the high cost of many new tests and the possibility that they may not provide informative results for some families. An individualized approach should consider factors such as a specific clinical course, physical findings, and family history to ensure that the most relevant tests are conducted.

To minimize misunderstandings and manage variable emotional reactions to genetic test results, healthcare providers should discuss inconclusive results and, if necessary, refer patients to providers with genetic expertise. This is especially important in cases where VUS is identified as these results can be difficult to interpret and may require additional evaluation to determine their impact on patient health and development.

Providers who present VUS results are responsible for providing families with appropriate information, support, and resources to facilitate their understanding of the results. Additionally, providers should consider whether further evaluations, such as parental or familial testing, imaging, or specialist referrals, could contribute to determining the pathogenicity of the variant and provide a more comprehensive understanding of the patient’s health status. By adopting an individualized approach to genetic testing and providing effective support to families throughout the process, healthcare providers can help ensure that patients receive the most accurate and informative results. The clinical practice is still ongoing.

## 5. Discovery and Clinical Application of Genetic Testing for DEEs

DEEs are a group of severe and early-onset epilepsies characterized by persistent seizures, developmental delay or regression, and poor prognosis. In 2017, the International League Against Epilepsy renamed the term “epileptic encephalopathy” as “developmental and epileptic encephalopathy” in consideration of the new insights into the genetic causes of epilepsy [36]. This change has been implemented by advances in the understanding of the genetic causes of epilepsy [37]. EEs are related to conditions whereby abundant epileptiform abnormalities and/or a high number of epileptic seizures contribute to cognitive regression [38]. This is typical in patients whose preceding function was normal or near normal. In such a condition, aggressive treatment should be considered, which may improve the outcome [39]. DEEs refer to conditions in which cognitive development and behavior are impaired independently of the onset of epilepsy, and epilepsy is characterized by a high frequency of seizures and abundant epileptiform abnormality [40]. Currently, 30–50% of DEEs are believed to have a genetic basis, and certain genes responsible for EEs have been linked to developmental delays and altered cognition [37,41]. Knowing the distinction between DEEs and EEs is crucial as it can inform the appropriate treatment approach. For instance, aggressive treatment of epileptic spasms may not improve cognitive disorders in many DEEs; therefore, harmful adverse events can be avoided with more tempered treatment. Thus, the development of accurate and efficient diagnostic protocols is crucial for determining the best treatment plan and optimizing the prognosis [39,42]. Cognitive delay occurs due to seizure and interictal seizure activity, as well as underlying neurophysiological processes [40]. The scientific literature provides clear examples, such as *SCN2A* and *KCNQ2*, where loss-of-function versus gain-of-function variants lead to differences in the clinical presentation [43]. Even when seizures cease early in life, individuals with these genetic conditions may experience developmental delay. In such cases, neurocognitive function may be improved through gene-targeted therapies [37,40]. Therefore, establishing a genetic diagnosis is crucial to guide precision medicine and prevent adverse outcomes. The identification of a specific underlying genetic variant can guide precision medicine to prevent the paradoxical aggravation of certain epilepsies. For example, the use of sodium channel blockers in children with DS may worsen their outcomes [44].

DEEs display genetic and phenotypic heterogeneity, and there are several options for genetic testing available. These range from gene panels that cover a few or hundreds of genes to ES, which investigates all ~20,000 genes [10]. Genetic testing options for DEEs are diverse and extensive. The most frequent causes of DEEs are sequence variants (30–40%) and chromosomal deletions or duplications (5–10%) [10,22]. Gene panels offer greater sequencing depth and lower costs than ES and genome sequencing (GS), but they examine only the specific genes included in the panel. Under such conditions, the benefit of a greater coverage depth is lost, but it reserves the possibility of future reanalysis to include the whole exome. ES, in contrast, provides a good sequencing depth at a lower cost but is limited to protein-coding regions. CNVs can be predicted using this method; however, a secondary method is required to plot breakpoints. The choice of the most suitable genetic test for DEEs depends on various factors, such as the age at which seizures first occur, condition severity, and patient insurance. Identification of DEE-associated genes is a rapidly evolving field, with many novel genes being discovered in recent years [10]. Many of these genes are involved in the regulation of neuronal ion channels, leading to over-excitability or reduction in inhibitory mechanisms [45,46]. Recent advances in genomic research have identified several genes that contribute to epileptic disorders beyond those coding for ion channels. These newly identified genes encode various proteins, such as chromatin remodelers, intracellular signaling molecules, metabolic enzymes, transcription factors, and mitochondrial complex genes [4,8,47]. Furthermore, de novo variants of *CACNA1E* have been recognized in individuals with DEEs [48,49]. Studies have also shown a correlation between DEEs and NDDs, with the discovery of novel genes associated with epilepsy syndromes, such as *NBEA*, *FBXO11*, and *SMARCC2* [50,51,52]. *NBEA,* in particular, is a long-standing candidate gene for NDDs and idiopathic autism [53].

Recessive genes may be uncommon but can be significant contributors to the underlying cause of DEEs, with many autosomal recessive epilepsies arising from inborn metabolic errors and cortical malformations [54]. In particular, genes involved in the biosynthesis and remodeling of glycosylphosphatidylinositol-anchored proteins, such as *PIGB*, reportedly cause autosomal recessive epilepsy [55]. 

The detection of pathogenic CNVs using CMA has demonstrated that CNVs contribute to 5–10% of childhood epilepsies, including DEEs [56,57,58]. Therefore, when the clinical presentation includes dysmorphism, congenital anomalies, intellectual disability, and other neuropsychiatric manifestations, CMA is the recommended first-line genetic test [59]. However, NGS is becoming a widely used diagnostic tool for the detection of CNVs. With the advent of ES and GS, detecting both single-nucleotide variations and CNVs are now feasible using a single test with an exome- or genome-wide approach [60].

Increasing evidence suggests that DNA methylation and histone modifications play causal or contributory roles in several medical conditions [61,62]. A recent study examined the role of de novo methylation changes in NDDs using methylation chips from 489 individuals [63]. Another study identified differentially methylated regions in two individuals with epilepsy and intellectual disabilities of unknown etiology [64].

### 5.1. Use of Genetic Testing for Precision Therapy Approaches

Precision medicine involved an individualized approach to medical care, emphasizing targeted treatment that is based on genetic tests, identification of biomarkers, and development of specific drugs [65]. Identifying the specific genetic cause of DEEs can inform the selection of the most appropriate treatment option [28]. Certain genetic mutations can provide therapeutic options that may be more effective for a particular patient. Based on current clinical evidence, some precision medicine strategies for epilepsy have been proposed [66,67]. However, early genetic testing is only the first step in the precise approach; functional testing is also required to determine pathogenicity and explore the fundamental functional impact, such as in the case of the *SCN1A* variant [68,69,70]. A comprehensive understanding of the molecular mechanisms underlying EEs is imperative for developing targeted and effective therapeutic solutions for these diseases at the earliest stage. For example, *SCN1A* is a commonly detected epilepsy gene in DS patients. The current focus of treatment for EEs is primarily on controlling seizures, but newer therapies, such as genetic treatments and antisense oligonucleotides, target specific causes, such as *SCN1A* channelopathy. A summary of drugs that have demonstrated recent progress in clinical studies, including gene therapy, is presented in Table 2. 

For example, treatment options for DS currently include stiripentol, fenfluramine, and cannabidiol, while research is exploring the potential of antisense oligonucleotides as a therapeutic approach [81,86]. Patients with SCN1A variants should avoid sodium channel blockers such as lamotrigine and carbamazepine since these drugs can worsen seizures. On the other hand, several medications have been shown to be effective in treating seizures in these patients. These include fenfluramine, cannabidiol, valproic acid, topiramate, clobazam, and stiripentol [71,72,73,87,88,89]. Other genetic epilepsies that can benefit from precision therapy include *KCNQ2* (carbamazepine, phenytoin) [90], *GLUT1* deficiency, *SLC2A1* or *CDKL5* (ketogenic diet) [91,92], *PCDH19* (clobazam) [93], mutations in *KCNQ2*, *SCN2A*, and *SCN8A* (ion sodium channel blockers), and mTORopathies (mTOR inhibitors) [82,94]. In addition, variants in genes encoding GABA receptors are a common cause of DEEs, and these variants are believed to reduce neuronal GABAergic activity through loss-of-function receptors. However, studies have shown that missense variants in *GABRB3* can lead to both gain-of-function and loss-of-function mutations, leading to unique clinical phenotypes [95]. Similarly, in patients with the Lennox–Gastaut syndrome, different point mutations in *CACNA1A* result in various clinical manifestations, with both gain-of-function and loss-of-function mutations associated with equally severe DEEs [96]. 

*SCN2A* mutations can also result in different functions at different times, causing early-onset DEEs owing to gain-of-function mutations and later-onset DEEs arising from loss-of-function mutations [97,98]. These observations demonstrate the compounding impact of both gene-related developmental impairment and epilepsy on development and cognition [37]. The use of existing treatments for epilepsy with specific genetic causes represents a significant advancement in this field [66]. The discovery of new treatments targeting genetic mutations in epilepsy will further enhance treatment options. Recently, advances have been made in the development of gene therapies for DEEs. Although clinical trials are still ongoing, these advances in gene therapy have opened new avenues for the treatment of DEEs, which have traditionally been challenging. 

Moreover, genetic testing may be important in guiding personalized treatment decisions for individuals with DEEs by providing insights into disease progression and treatment response. As our understanding of the genetics of epilepsy continues to increase, genetic testing is likely to be increasingly important in treatment decisions. Genetic counseling can also provide parents of children with DEEs with information on preimplantation or prenatal genetic diagnostics. These tests can offer choices and possibilities for prevention, allowing parents to make informed decisions about their options for family planning [99]. Therefore, it is crucial to consider genetic testing along with other clinical factors, such as symptoms and medication responses, to enable informed treatment decisions.

### 5.2. Limitations and Challenges of Genetic Testing in DEEs

Several limitations and challenges of genetic testing persist as follows:Limited availability of testing: In many regions, access to genetic testing is limited, particularly comprehensive testing, which may require specialized facilities and expertise.Interpreting test results: Interpreting genetic test results can be challenging, particularly when multiple genes are involved, and not all genetic variations have clear implications for the diagnosis or treatment of DEEs.False positive results: False positive results can occur, leading to anxiety and further testing of patients and families, which may result in the misallocation of limited healthcare resources.Limitations of current genetic tests: Current genetic tests are limited by their inability to detect some mutations and structural variations, and their potential for missing disease-causing mutations.Cost: The cost of genetic testing can be a barrier for some families, especially if insurance excludes it.Ethical concerns: Genetic testing raises ethical concerns, including the potential for discrimination based on genetic information, privacy, and security of genetic information and the potential for emotional distress caused by the knowledge of genetic risk.Difficulty extrapolating test results: Finally, it is difficult to extrapolate test results to guide therapeutic strategies. For example, even if a genetic mutation is identified, it may be unclear how it contributes to the development of DEEs or how it should inform therapeutic strategies.

Therefore, genetic testing for DEEs should be approached with caution, considering the limitations and challenges discussed above and with the support of an expert healthcare professional.

## 6. Conclusions

Neurodevelopmental disorders are frequently occurring conditions, and many of them are caused by uncommon copy number and exon sequence variations. These variations can be identified by genetic testing, which helps in the diagnosis and management of such disorders. Children with severe developmental delays and epilepsy mostly have poor prognoses. Identifying the genetic cause of a disease can enable clinicians to provide more precise prognostication and counseling on the risk of recurrence, as well as to prevent and treat medical comorbidities. In addition, It can also direct patients and families toward appropriate resources, both locally and internationally, and refine treatment options where possible. Among the multiple genetic tests available today, NGS is considered a valuable and dependable diagnostic tool for detecting gene mutations in children with epilepsy and developmental disabilities. In particular, this cost-effective method shortens the time from seizure onset to genetic diagnosis. By comprehending the gene families involved in epilepsy, we can gain insight into the intricate neuropathogenic pathways underlying the disease and develop more effective and precise therapeutic approaches. Therefore, we highlight the possibility of genetic testing as a first-line investigation in children with DEEs. 

Advances in epilepsy genomics have provided significant benefits in the field of DEEs. With the availability of complete gene panels, exomes, and genomes, genetic testing has become an important tool for improving diagnostic rates and understanding the underlying disease processes. Their use has also opened new avenues for personalized medicine, allowing for more precise therapeutic strategies based on a patient’s unique genetic profile. However, much work remains to be done in this regard. Although current testing methods have been successful in identifying causal mutations, many limitations and challenges remain to be overcome, particularly with regard to understanding the complex interactions between genetic and environmental factors. Therefore, continued research on the genetics of DEEs is critical. By exploring new technologies and refining existing methods, scientists can gain a deeper understanding of the underlying causes of these debilitating conditions, leading to better diagnostics, treatments, and, ultimately, improved outcomes for affected individuals and their families.

Further research is suggested, including 1. new techniques and algorithms to enhance the accuracy of genetic testing for DEEs and identify more disease-causing mutations, 2. using genetic testing results to inform further studies aimed at understanding the biological basis of DEEs, and 3. developing new therapies based on genetic testing results to improve the outcomes of patients with DEEs.

## Figures and Tables

**Figure 1 children-10-00556-f001:**
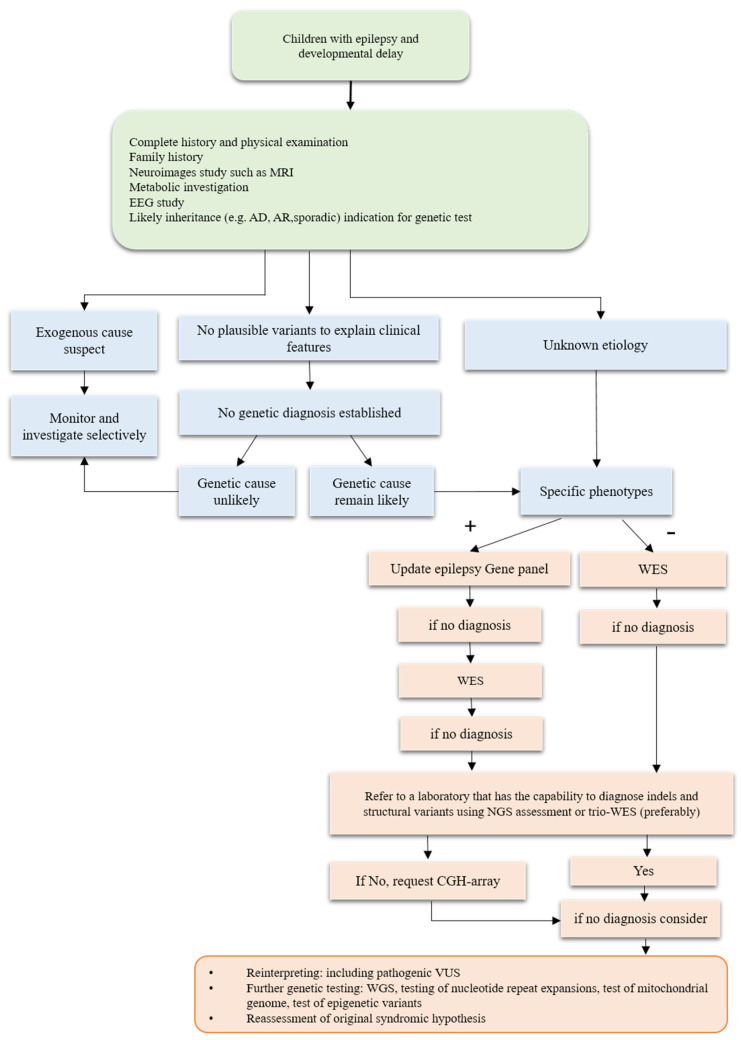
Flowchart illustrating the investigation process for developmental and epileptic encephalopathies. AD, autosomal dominant; AR, autosomal recessive; EEG, electroencephalography; MRI, magnetic resonance imaging; VUS, variant of unknown significance; WES, whole-genome sequencing.

**Table 1 children-10-00556-t001:** Characteristics of gene panels versus WES.

	Gene Panel	WES
Cost	Low	High
Diagnostic rate	Usually 15–48%, but may be 0.8% at lowest	25–44%
Cost	Low	High
Time	Rapid	Slow
Advantages	Rapid Cost-effective	Covers entire coding sequence Trio exome sequencing can discover de novo variants Allows for further reanalysis Can potentially be used to detect CNV
Disadvantages	Only test the genes on the panel The result may quite variable depend on the genes on the panel May miss as yet unknown disease causing genes Fewer VUS than WES	Incidental findings Interpretation of multiple VUS may be necessary May identify carrier status or non-paternity Unable to detect deep intronic mutations, structural rearrangements, or large deletions/duplications

CNV, copy number variant; VUS, variants of uncertain significance; WES, whole-exome sequencing.

**Table 2 children-10-00556-t002:** Diseases and drugs as subjects of recent advances in patients with DEEs.

Disease	Pathophysiological Background	Drug	Mechanisms of Action	Reference
Dravet syndrome	Haploinsufficiency of the voltage-gated sodium channel α subunit NaV1.1	Stiripentol	Allosteric modulator of benzodiazepine-sensitive/benzodiazepine-insensitive GABAA receptor; activating ATP-sensitive potassium channels	[71]
		Cannabidiol	*GPR55*, *TRPV1*, and adenosine reuptake	[72,73]
		Soticlestat	Brain specific cholesterol 24-hydroxylase inhibitor; dose-dependently reduced plasma 24*S*-hydroxycholesterol; decreases excitability	[74,75,76]
		Fenfluramine	Serotonin (5-HT) release; increases serotonergic signaling; more specific of 5-HT 1D and 5-HT 2C receptors	[77]
		dCas9-mediated *Scn1a* gene activation system (murine model)	Stimulates *Scn1a* transcription	[78]
*KCNQ2* mutation	*KCNQ2* mutation	Ezogabine/retigabine	Specific activator of voltage- gated potassium Kv7.2/7.3 channels; decreases excitatory neurotransmission	[79]
	*KCNQ2* loss-of-function as a more precise indication; early infantile epileptic encephalopathy type 7 (BFNS)	XEN1101	Selective potassium channel opener; decreases excitatory neurotransmission	[80,81]
TSC	Mutations in *TSC1* or *TSC2*	Everolimus	mTOR inhibitor; mutations lead to excessive activation of mTOR signaling pathway, abnormal cell differentiation, altered plasticity, and inflammatory signaling	[82]
SCN8A mutation	Gain-of-function mutations encoding the Nav1.6 channel (EIEE13)	NBI 921352 (XEN901)	Selective inhibitor of voltage-gated sodium channel subtype Nav1.6, could address the cause of this condition	[81,83]
DEEs	De novo variants in the gene encoding dynamin-1 (*DNM1*)	RNAi-based gene therapy (murine model)	*Dnm1*-targeted therapeutic microRNA delivered by a self-complementary adeno-associated virus vector	[84]
STXBP1-encephalopathy	Mutations in *STXBP1*		Specific protein–protein interaction inhibition and gene therapy	[85]

BFNS, benign familial neonatal seizures; DEEs, developmental and epileptic encephalopathies; TSC, tuberous sclerosis complex.

## Data Availability

Not applicable.

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
