# Peer review of "Genetic Testing in Children with Developmental and Epileptic Encephalopathies: A Review of Advances in Epilepsy Genomics"

_children, 2023, doi:10.3390/children10030556_

Round 1

Reviewer 1 Report

This review articels summarizes some recent advances in genetic testing for DEE. There are some concerns that the authors should address:

Major:

1.       Section C in introduction is redundant. There is no important information other than the point that genetic testing is important. This whole section can be removed.

2.       Line 186, the authors mentioned CMA without really giving an introduction to what it is. The readers outside the field may be confused by it with WES, WGS and gene panels. The authors should give a brief introduction to CMA and tell its differences from the other methods.

3.       Line 224, the authors should make more clear what the distinctions are between DEE and EE and how this affects clinical applications.

4.       Figure 1 is not completely shown in the pdf. Please make sure the whole figure can be shown in the pdf.

Minor:

1.       First sentence in abstract, “advanced” should be advances. But still there are two advances in the same sentence.

2.       Line 63, “Epilepsy is now” should be changed to “It is now”

3.       Line 152, “The diagnostic yield of gene panels is also depend on the population been tested. Even lowest to 0.8%” should be changed to “The diagnostic yield of gene panels is also dependent on the population being tested, which can be as low as 0.8%”

4.       Line 273 and 274. Two sentences are redundant.

5.       Line 214, “The clinical trials are still going on”

6.       Table 1, Diagnostic rate for Gene panel should be “But may be 0.8% at lowest”

7.       In general, language should be more concise and less redundant.

8.       Please make fonts consistent.

Author Response

Please see the attachment. We appreciate your valuable feedback and suggestions.We had sent this manuscript to a professional English editing service to ensure that the language is concise and not redundant.

Reviewer 1

Comments and Suggestions for Authors

This review articles summarizes some recent advances in genetic testing for DEE. There are some concerns that the authors should address:

Major:

  1. Section C in introduction is redundant. There is no important information other than the point that genetic testing is important. This whole section can be removed.

Reply: Thank you for your feedback regarding Section C in the introduction. After careful consideration, I have decided to remove this section as it was redundant and did not provide any additional important information beyond the importance of genetic testing. I appreciate your valuable input and thank you for helping me improve the manuscript.

  1. Line 186, the authors mentioned CMA without really giving an introduction to what it is. The readers outside the field may be confused by it with WES, WGS and gene panels. The authors should give a brief introduction to CMA and tell its differences from the other methods.

Reply: We appreciate your suggestion and have revised the manuscript accordingly. In line 163-171, we have added more content about CMA and provided a brief introduction to the method. Additionally, we have explained its differences from other genetic testing methods such as WES, WGS, and gene panels to help readers outside the field understand the distinctions between them.

  1. Line 224, the authors should make more clear what the distinctions are between DEE and EE and how this affects clinical applications.

Reply: Thank you for your valuable feedback. We have taken your suggestion and added more content in lines 211-218 to clarify the distinction between DEE and EE. We have also explained how this distinction affects clinical applications.

  1. Figure 1 is not completely shown in the pdf. Please make sure the whole figure can be shown in the pdf.

Reply: Thank you for bringing to our attention the issue with Figure 1 not being completely shown in the pdf. We have taken steps to ensure that the entire figure is now visible in the pdf.

Minor:

  1. First sentence in abstract, “advanced” should be advances. But still there are two advances in the same sentence.

Reply: We have modified the first sentence of the abstract to read: Advances in disease-related gene discovery have led to tremendous innovations in epilepsy genetics.

  1. Line 63, “Epilepsy is now” should be changed to “It is now”

Reply: Thank you. We had made the necessary revisions in Line 65-68

  1. Line 152, “The diagnostic yield of gene panels is also depend on the population been tested. Even lowest to 0.8%” should be changed to “The diagnostic yield of gene panels is also dependent on the population being tested, which can be as low as 0.8%”

Reply: Thank you. We have addressed your concern regarding the sentence in line 152. We have revised the sentence now in Line 126

  1. Line 273 and 274. Two sentences are redundant.

Reply: We had revised the redundant sentences now line 269-270

  1. Line 214, “The clinical trials are still going on”

Reply: Thank you for your valuable feedback. We already add this opinion lin line 204

  1. Table 1, Diagnostic rate for Gene panel should be “But may be 0.8% at lowest”

Reply: Thank you for your comments. Regarding your comment about Table 1, we have made the necessary correction and updated the diagnostic rate for Gene panel to "But may be as low as 0.8%".

  1. In general, language should be more concise and less redundant.

Reply : Thank you for your feedback. We have taken it into consideration and have already sent the manuscript to a professional English editing service to ensure that the language is concise and not redundant. We appreciate your input and hope that you will find the revised manuscript to be of high quality.

  1. Please make fonts consistent.

Reply: Thank you for your valuable feedback on our manuscript. We have addressed your comment regarding font consistency and made the necessary adjustments throughout the manuscript.

Reviewer 2 Report

The authors provide a review of recent advances in genetic research into causes of epileptic encephalopathies. However, there are many repititions in the text, making the manuscript too long and not concise enough.

The authors state that a better knowledge of the causes of epileptic encephalopathies helps to find precise therapies. Here I miss an important other aspect of the advances in genetic analyses. That is the possibility of pre-implantation or prenatal genetic diagnostics of developmental and epileptic encephalopathies. With the possibility of pre-implantation or prenatal diagnostics choises arise for the expectant parents and possibilities of prevention. 

Author Response

 We appreciate your valuable feedback and suggestions. We also had sent the manuscript to a professional English editing service to ensure that the language is concise and not redundant. The reply , please see the attachment.

Reviewer

The authors provide a review of recent advances in genetic research into causes of epileptic encephalopathies. However, there are many repititions in the text, making the manuscript too long and not concise enough.

The authors state that a better knowledge of the causes of epileptic encephalopathies helps to find precise therapies. Here I miss an important other aspect of the advances in genetic analyses. That is the possibility of pre-implantation or prenatal genetic diagnostics of developmental and epileptic encephalopathies. With the possibility of pre-implantation or prenatal diagnostics choises arise for the expectant parents and possibilities of prevention. 

Besides, genetic counselling can also provide parents with children who have DEEs with information on pre-implantation or prenatal genetic diagnostics. These tests can offer choices and possibilities for prevention, allowing parents to make informed decisions about their options for family planning (line 331-334)

Reply :

  1. Thank you for your feedback. We have taken it into consideration and have already sent the manuscript to a professional English editing service to ensure that the language is concise and not redundant. We appreciate your input and hope that you will find the revised manuscript to be of high quality.
  2. Thank you for your comments and suggestions. We would like to inform you that we have added a section on pre-implantation and prenatal genetic diagnostics in the manuscript (lines 328-331).We hope that this addition will further enhance the usefulness of our manuscript. Once again, we appreciate your valuable feedback and suggestions.

Round 2

Reviewer 1 Report

1. This review focuses on DEE but the background only talks about epilepsy and EE without clearly defining DEE. It is nice that the authors elaborated on the differences between the defnitions of DEE and EE in line 205-217. But it would make more sense if this can be moved to the background/intro part. Of course minor rewriting may be necessary.

2. Figure 1, "Genetic cause unlikely selectively" sounds weird. Is "selectively" extra?

3. Still figure 1, there is no arrow after "WES- -> if no diagnosis". I assume this should also lead to the "refer to a laboratory ..." block?

4. Line 236, "area" should be "are".

Reviewer 2 Report

No other comments 

Author Response

Thank you for your valuable feedback.